# Bioturbation in the hadal zone

Jussi Hovikoski [1] ✉, Joonas J. Virtasalo [2], Andreas Wetzel [3], Mishelle Muthre [4], Michael Strasser [4], Jean-Noel Proust [5] & Ken Ikehara [6]

The hadal zone, >6 km deep, remains one of the least understood ecosystems on Earth. We address bioturbational structures in sediment cores from depths exceeding 7.5 km, collected during the IODP Expedition 386 in the Japan Trench. Micro-CT imaging on 20 core sections allowed to identify biogenic sedimentary structures (incipient trace fossils) and their colonization successions within gravity flow deposits. Their frequency, and consequent changes in substrate consistency, oxygenation and organic matter delivery and remineralization controlled the endobenthic colonization. The gravity-flow beds show recurring bioturbation successions: The initial colonization is characterized by deposit-feeding structures such as *Phycosiphon, Nereites* and *Artichnus* generating typically 20 cm thick intensively bioturbated fabrics. The final colonization stage comprises slender spiral, lobate and deeply penetrating straight and ramifying burrow systems such as *Gyrolithes, Pilichnus* and *Trichichnus*, interpreted to include burrows of microbe farming and chemosymbiotic invertebrates. The main factor precluding colonization is soupy substrate. Organic matter degradation and post-event upward expansion of the anoxic zone drive the change from deposit feeding to microbe-dependent feeding strategies.

The hadal zone represents the deepest part of the world's ocean, 6–11 km deep, being mostly confined into subduction-related trenches along active convergent margins[1]. The hadal trenches are narrow, elongate, isolated basins, bounded by the abyssal plains, and characterized by steep slopes, low temperature and immense water pressure[2]. Zoobenthic studies stated that the diversity, abundance and biomass of metazoans generally decrease with depth and reach particularly low levels in the abyssal plains[3]. However, recent advances in understanding the hadal environment suggest that it can host species diversity and habitat heterogeneity comparable to that of shallow-water environments[1,4,5]. Indeed, sampling campaigns in hadal trenches have revealed a diverse array of metazoans, comprising primarily benthic fauna, such as holothurians, polychaetes, bivalves, isopods, actinians, amphipods and gastropods as well as bottom-living fish[6]. In addition, microbial activity and benthic carbon mineralization rates in hadal trenches are high compared to the adjacent abyssal plains[7–10]. The increased biodiversity and carbon mineralization rates in the hadal

trenches result from the episodic delivery of organic matter via gravity flows[8,11–14], which provide a supply of nutritious material to support a fauna in time-stable environment.

An outstanding challenge for understanding hadal benthic habitats and their modulating factors is the lack of knowledge regarding bioturbation, which is a key ecosystem engineering process that mixes, irrigates, and oxygenates sediments and affects sediment properties such as texture, porosity and permeability[15,16]. The influence of bioturbation thus extends to processes such as nutrient cycling, early diagenesis and remineralization versus preservation of organic matter[17,18]. For instance, burrowing irregular echinoids and holothurians can have a critical impact on nutrient fluxes and productivity and hence far reaching influences on ecosystem functioning[15,19].

In addition to the ecosystem-engineering role of burrowing animals, bioturbated sediments sensitively record abiotic processes that operate in the system, for example the position of the Redox Potential Discontinuity (RPD)[20]. By studying biogenic sedimentary structures, in

[1]Information Solutions, Geological Survey of Finland (GTK), Espoo, Finland. [2]Marine Geology, Geological Survey of Finland (GTK), Espoo, Finland. [3]Departement Umweltwissenschaften – Geologie, Universität Basel, Basel, Switzerland. [4]Department of Geology, University of Innsbruck, Innsbruck, Austria. [5]Géosciences, CNRS, University of Rennes, Rennes, France. [6]Geological Survey of Japan, National Institute of Advanced Industrial Science and Technology (AIST), Tsukuba Central 7, Tsukuba, Ibaraki, Japan. ✉e-mail: jussi.hovikoski@gtk.fi

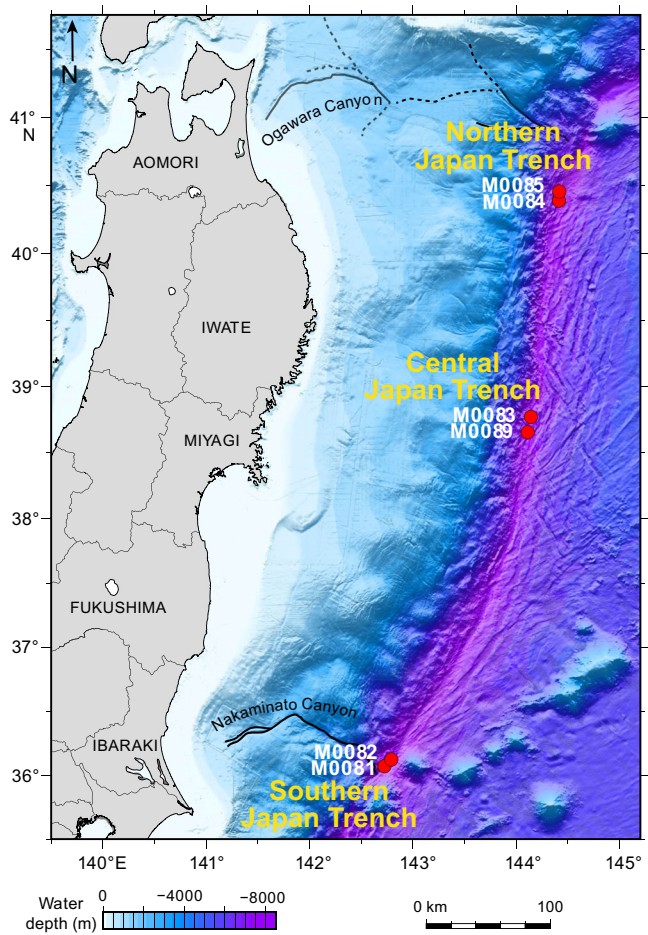

**Fig. 1 | Map of the Japan Trench (modified after Kioka et al. 2019[12]).** Red dots indicate site locations of the studied cores. Black bold and dashed lines mark the Nakaminato and Ogawara canyon in the south and north, respectively.

particular trace fossils and their bioturbational fabrics, we can gain information regarding ecological factors such as sedimentation rate, event frequency, changes in oxygenation and substrate consistency[21,22]. Unfortunately, in addition to the lack of documentation of modern bioturbation in the hadal zone, there are no fossil examples of such sediments in the rock record to author's knowledge, probably due to the limited preservation potential of deposits in tectonically-active subduction zones. As a result, the hadal zone has remained the only bathymetric zone for which no ichnological data is available.

The International Ocean Discovery Program (IODP) Expedition 386 successfully collected 29 giant piston cores of upper Pleistocene to Holocene sediments from the Japan Trench in 2021 (water depth > 7.5 km; Fig. 1)[23,24]. We applied micro-CT imaging on 20 selected whole-round core sections (each 17–55 cm long), which were retrieved from (1) relatively more expanded sections within the main depocenters experiencing the highest sedimentation rates and from (2) relatively more condensed sections in a relative bathymetric high in the trench-basins of Southern, Central and Northern Japan Trench (Supplementary Note 1). The data set was supplemented with grain size and C/N analyses ($n = 48$) from the studied whole-round samples, which were further combined with previously published analyses from the same cores ($n = 126$)[24]. The purpose of this study is to document biogenic sedimentary structures and colonization successions in the selected cores from the hadal zone. The ichnological data are combined with sedimentological descriptions and depositional-process interpretation. Relationships of burrowing depth, trace density and

diameter and sediment properties between trench basins as well as between expanded and condensed sections of the basins are assessed statistically. The results show that the burrowing activity is strongly controlled by event sedimentation and its frequency, and follows predictable ethological trends, driven by changes in substrate consistency, organic matter utilization and (pore)water oxygenation. These findings form the basis for a sedimentological-ichnological model for event-bed colonization in the hadal zone.

## Results

A synthesis of main sedimentological and ichnological properties is presented below. Full description and photo-documentation of burrow morphologies, sedimentary facies and core sections are provided in Supplementary Notes 2–4.

### Sedimentology

The deposits are divided into three facies (F1–F3), with F1 and F2 being further subdivided into four and three sub-facies, respectively (F1A–D, F2A–C; Supplementary Notes 2 and 3). Typically, the deposits are arranged into sharp-based, generally upward fining (base-graded) beds, 10–50 cm thick, comprising (sand)-silt-clay grain sizes (Fig. 2A). The base or lower part of the beds commonly exhibits ripple cross-laminated silt (F3), which can grade upward into, or be interbedded with, silt-clay interlamination (F2A and B) and convoluted mud (F2C). The silt-clay interlamination can be horizontal and parallel (F2A) or show convergent laminae, foresets, laminae pinch outs and down-lapping laminae contacts (F2B; Fig. 2B–D). The upper part of a bed is characterized by a colonization surface formed during slow pelagic sedimentation (F1A) having below bioturbated mud (F1B), macroscopically ungraded structureless mud (F1C) and/ or graded mud (F1D). Soft-sediment deformation structures like loading structures may occur throughout a bed and are commonly associated with the structureless mud facies (F1C); an illustrative example is a pseudonodular silt fabric, forming detached ball-and-pillow structures that descend as deep as 20 cm into the underlying structureless mud (Fig. 2E).

The sediments are interpreted to have been deposited from pulsed, but generally decelerating gravity flows. The lower part of an event bed typically displays ripple cross-laminated (sandy) silt (F3) and clay-prone mud ripples (F2), which indicate deposition from traction currents, transporting both non-cohesive sand and silt as well as cohesive clay particles. In addition to well-defined mud-ripple cross-lamination, recurrently occurring convergent and down-lapping laminae indicate the presence of mud-floccule ripples, formed by traction under turbulent, low-density sediment flows (Fig. 2D)[25,26]. The transition to the overlying or interbedded structureless mud (F1C) is locally associated with decreasing ripple size and foreset angle and/or increasing clay content, which point to suppressed turbulence by increasing sediment concentration and cohesive mud content in response to flow deceleration (Fig. 2C)[27]. This interpretation is supported by intra-facies (laminae- or bed-bound) convolute lamination and micro-scale loading structures, indicating high water content of the sediments. The overlying structureless mud (F1C) appears ungraded and shows locally evidence for soup-ground consistency and rapid sedimentation, such as loading structures, pseudonodular fabric, and rare deformed escape burrows (Fig. 2E). This facies appears to include both deposition of fluid mud and from settling suspension, which cannot be clearly distinguished (so-called "densite mud")[26]. Locally present normally graded mud bands point to differential settling of silt and clay from a suspension cloud.

Finally, hemipelagic-pelagic deposits, formed by settling from background dilute suspensions, cannot be recognized by using the available sedimentological data alone. The preservation of hemipelagic-pelagic laminae can be limited by erosion generated by the emplacement of the overlying event bed. However, the decline of

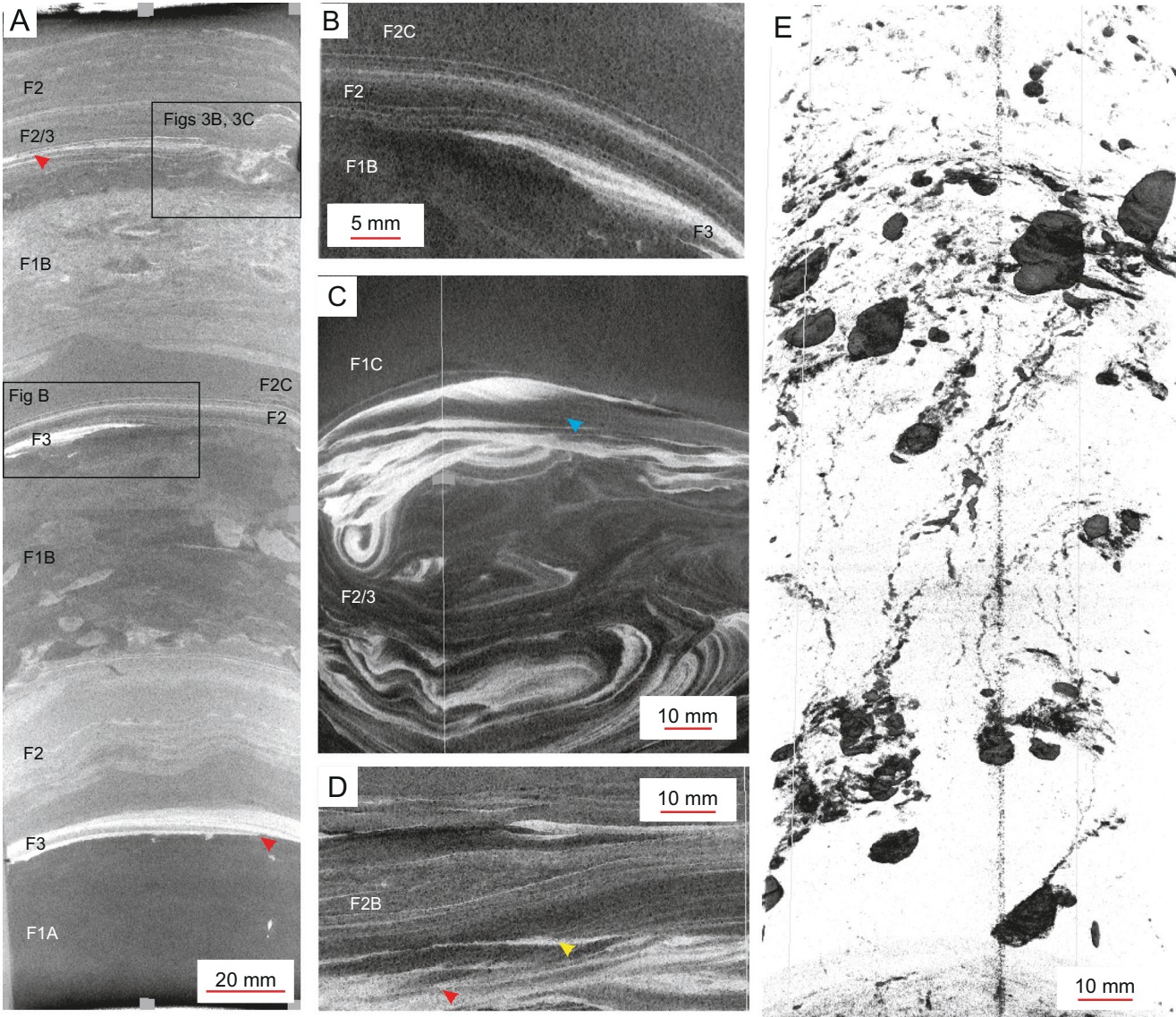

**Fig. 2 | Main sedimentological characteristics of the studied cores. A** Stacked, sharp-based and base-graded event beds comprising recurring facies successions: Base of the successions is formed by ripple cross-laminated silt (F3), which is followed by laminated clay and silt (F2). These deposits are, in turn, overlain by structureless mud (F2C) and/or bioturbated mud (F1B). Red arrow – low-angle ripple cross-lamination. See text for explanation. Core section M0081D-1H-6WR2, 0–30 cm, Southern Japan Trench expanded section. **B** Close up of the base of a gravity-flow bed, showing ripple cross-laminated silt (F3) that grades upward into laminated clay and silt (F2). The lamination starts with subtle, lateral thickness variation with laminae terminations, followed by parallel lamination and transition to structureless mud (F2C). **C** Slump-folding and loading structures in clay-silt heteroliths. The uppermost ripple cross-lamination grades laterally into laminated mud (blue arrow). Soft-sediment deformation is restricted to a specific bed interval pointing to syn-sedimentary origin and high initial water content of the sediment. Core section M0082C-1P-1WR6, 63–72 cm, Southern Japan Trench condensed section. **D** Laminated to cross laminated mud with foresets (red arrow) and down-lapping laminae terminations (yellow arrow). Core section M0082C-1P-1WR6, 74–80 cm, Southern Japan Trench condensed section. **E** High-contrast 3D-view to pseudo-nodular silt/sand, sunken into structureless mud. Core section M0083F-1H-10WR16, 0–21 cm, Central Japan Trench expanded section. Note that the core margins have bent downwards due to coring process.

densite-mud deposition and a shift to nearly clear water conditions can be inferred from the preserved trace assemblage that is typical of decreasing sedimentation rate, increasing sediment consistency and development of a distinct colonization surface, pointing to a near cessation of sedimentation (F1A).

## Carbon and nitrogen data

The total organic carbon (TOC) contents of the sediment ranged between 0.25 and 2.81 wt.% (mean = 1.13 wt.%, $n = 48$) and the atomic C/N ratios ranged between 6.3 and 9.6 (mean=7.9, $n = 48$)[28]. The C/N ratios were similar to previously reported values from the Japan Trench and indicate that the sedimentary organic matter is mainly composed of marine organic matter[29]. The samples with the highest TOC contents typically had also the highest C/N ratios, which approach the C/N values documented for surface mud on the trench upper slope (8.5–9.4)[30]. The between-facies differences are small, but the highest TOC contents and C/N ratios were measured in F2. Apparently, the silt-clay laminae (F2), contain a larger proportion of organic-rich mud that originated from the upper slope. However, most of the sediments are redeposited mud with slightly more microbially degraded organic matter, originating from seismically-triggered remobilization of surficial sediment on the lower trench slope[12,29–31].

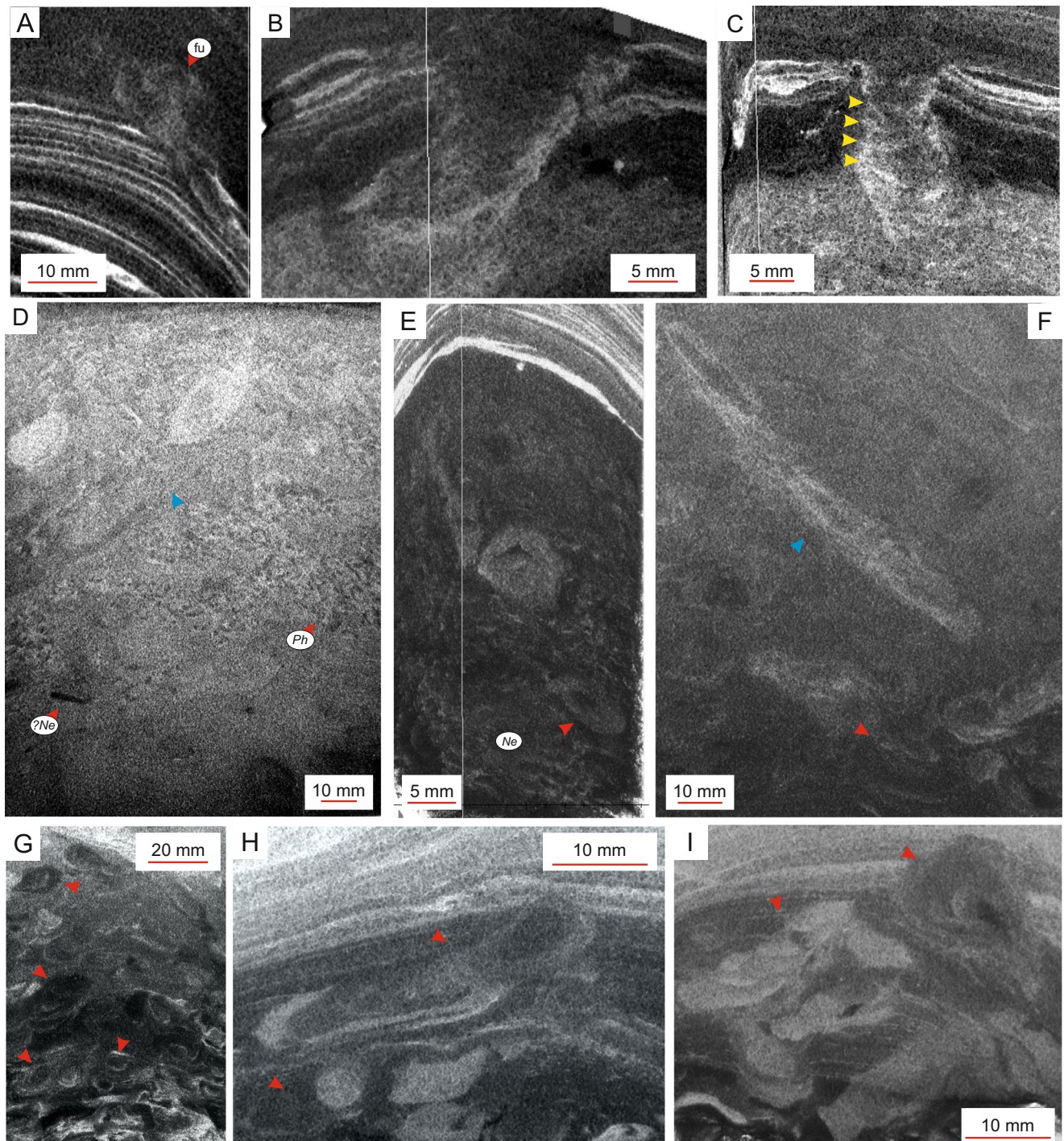

**Fig. 3 | Traces assigned to early post-event suites. White vertical lines mark the margin between image surfaces oriented in right angle. A** Fugichnia (fu) escape burrow extending upwards from clay-silt lamination (F2) to structureless mud (F1C). Core section M0085D-1H-3WR27, 92–94 cm, Northern Japan Trench condensed section. **B**, **C** Oblique side- and frontal-view of *Rhizocorallium*, respectively. Yellow arrows indicate spreite. Core section M0081D-1H-6WR2, 4.5–6.5 cm, Southern Japan Trench expanded section. **D**–**I** Various expressions of Rosselichnid traces. Red arrows point to potential *Artichnus*, whereas blue arrows indicate sub-vertical elements. In (**D**, **E**), Rosselichnid traces cross-cut a fabric dominated by *Phycosiphon* (*Ph*) and *Nereites* (*Ne*), respectively. Core sections: (**D**) M0089D-1H-5WR36, 29.5–40.5 cm, Central Japan Trench condensed section; (**E**) M0085D-1H-3WR27, 95–100 cm, Northern Japan Trench condensed section; (**F**, **G**) M0082D-1H-2WR7, 80–92 cm and 90–100 cm, respectively, Southern Japan Trench condensed section; **H** and **I**) M0081F-1H-7WR4, 96–101 cm, Southern Japan Trench expanded section. Note that the core margins have bent downwards due to coring process.

## Bioturbational structures

More than ten traces and/or burrows with recurrent morphology were identified (Supplementary Note 4). The described traces can be divided into two main groups (suites) based on (1) location relative to the colonization surface, (2) position within the event-bed facies, and (3) cross-cutting relationships. As found in the rock record, traces produced after a depositional event, the so-called "post-depositional" traces, can be distinguished from those produced prior to and preserved below an event bed, the so-called "pre-depositional" traces[32].

## Post-depositional opportunistic colonization

The basal parts of gravity-flow beds typically lack identifiable traces but may exhibit rare biodeformational and escape structures (Fig. 3A) in laminated mud and structureless mud (F2; Bioturbation Index (BI) 0–1[33]). A single *Rhizocorallium* isp. descends from a heterolithic laminated mud bed (Fig. 3B, C, Supplementary Movie 1).

Colonization of the upper part of a gravity-flow bed started immediately after the main, traction-transport phase of the gravity flow, from the at-that-time actual seafloor (Fig. 3). This trace suite comprises burrows of opportunistic animals (F1B) and is present in all trench basins; it commonly includes the deposit-feeding traces *Phycosiphon* and/or *Nereites* (Fig. 3D, E). Their producers exploited the freshly deposited, still oxygenated organic-rich mud. This trace suite does not, however, occur in gravity-flow beds, which are dominated by structureless ungraded densite mud (F1C). In addition to these feeding structures, horizontally to sub-vertically oriented burrows with a variably concentric or eccentric lamination enveloping a central tube are very commonly present (Fig. 3D–I). These burrows are assigned to the Rosselichnidae ichnofamily[34]. Common burrow morphologies include horizontal shoe-shaped traces, which are 7–12 mm in diameter and display a central, finer-grained fill surrounded by irregular, concentric laminae (Fig. 3D, G, H, I). Occasionally, a package of laminae at the base of a burrow appears as a retrusive spreite. The central fill is best visible where the short vertical shaft turns horizontal. Some of these Rosellichnid burrows show cross-section typical of *Artichnus*, which is considered to be produced by holothurians[35,36]. The Rosselichnid traces either cross-cut *Phycosiphon* and *Nereites* or constitute a monogeneric to low-diversity, intensively mottled fabric typically 20 cm thick.

Subordinately occurring Rosellichnid burrows are sub-vertical, bow-shaped and show variable lamination (7–10 mm diameter; Fig. 3D, F). The burrows can occur solitarily (cf. *Cylindrichnus*) or form overlapping or closely spaced bundles, bearing locally transitional morphologies to *Phycodes* (see Supplementary Note 4).

The post-depositional burrowers commonly bioturbated the sediment intensely and their traces overprint the primary sedimentary texture completely (BI 5–6), but in a few cases only gently (BI 1) (F1B). The bioturbated zone ranges from 5 to more than 50 cm in thickness. The burrowing depth appears to have been largest right after the gravity-flow bed emplacement when the porewater was still oxygenated. Sedimentary organic matter degradation consumes porewater oxygen and in the order of a few years the RPD moves upward[37,38]. It restricts the vertical penetration of burrowing organisms having an open connection to the seafloor. In cases of incomplete bioturbation, primary laminae and normal grading remain locally visible in the host sediment (F1B).

## Re-establishment of prevalent equilibrium burrowing fauna

The colonization by animals, bioturbating the sediment during the long periods between gravity-flow events, is characterized by a low-density and low-diversity trace suite that cross-cuts the early post-depositional suite. Typical of this equilibrium trace suite are burrows that penetrate downward from a colonization surface on the top of gravity-flow deposits (F1A). Burrowing depth is up to 50 cm. The traces are undeformed, slender with sub-mm to a few mm diameter and commonly passively filled, implying that they were originally open tubes. The sharp burrow boundaries indicate that they were emplaced in somewhat dewatered substrate (gravity flow-derived mud). The gravity-flow-related turbid conditions, therefore, must have been followed by a period of clear water or very dilute slowly accreting hemipelagic deposition that allowed the dewatering[39] and development of a single colonization surface. The burrow infills are typically denser than the sediment matrix and are well visible in high-contrast CT-scans despite of their minute diameter (Fig. 4).

Main traces include the following, listed broadly according to tiering depth:

1. Vertical to sub-vertical helicoidal burrows (cf. *Gyrolithes lorcaensis*), 0.5–1 mm in diameter, with sub-ordinate, short, blind-ending branches (Fig. 4A, B, Supplementary Movie 2) occur in all three trench basins. They penetrate as deep as 1–2 cm. Burrows of similar morphology with pyritic infill have been interpreted to record farming (agrichnial) behavior of the burrow producers that cultivated microbes. The helicoidal morphology may provide an enlarged tube surface area against surrounding sediment, characterized by contrasting redox conditions, facilitating microbial growth[40,41] while water circulation in the tube is optimal[42].

2. Lobate burrows are up to 5 cm high and show a crescentic or massive-appearing fill structure (Fig. 4G, H, Supplementary Movie 3). The lobes are narrow reaching a width of 13 mm, while their thickness is 1–3 mm. Moreover, the lobes can show narrow appendixes or branches, and can be attached to a (sub-)vertical helicoidal shaft 3 cm long and 1 mm in diameter. Such burrows are locally present in the relatively more condensed basin margin deposits in Southern Japan Trench. Burrowing depth ranges from 3 to 7 cm.

   Of the established ichnogenera, the traces bear mostly similarities with *Zoophycos*, although the miniature size is unusual. *Zoophycos* burrows are interpreted to reflect various ethological functions, including deposit feeding[43], inverse conveyor activity[44], cache, refuse dump, cultivation of symbiotic microorganisms[45], and combined detritus feeding and cache[46,47].

3. Extensive burrow systems of 1–2 mm tube diameter, penetrating 5–10 cm deep with mostly 1st and 2nd order, irregular and variably oriented branching that form root-like, several cm-long extensions occur in Southern and Central Japan Trench (Fig. 4D, E; Supplementary Movie 4). They are classified as *Pilichnus* isp., a genus belonging to the *Chondrites* group[48,49]. Locally, the deepest burrow extensions in the here documented burrows may show narrow sub-vertical helicoidal segments (Fig. 4E). The branching tube system is connected to the seafloor by a few-mm wide, straight to J-shaped, 2–4 cm long, vertical to inclined shaft (Fig. 4D, F). These traces resemble the burrows of thyasirid bivalves, which construct extensive ramifying burrow systems[50–52]. Their irregular *Chondrites*-like burrow morphology with simple 1st and 2nd order branching has been linked to thyasirids that live in symbiosis with chemoautotrophic, sulfur-oxidizing bacteria[51,53] The deep-penetrating, branching tubes provide access to reduced sulphur, whereas the vertical, descending tubes may function as inhalant tubes, connecting the burrow system to the seafloor. Similar burrows can also be formed by some asymbiotic thyasirids, which cultivate chemosynthetic microbes along the tube walls[52]. Thyasirid bivalves are known to occur in the Japan Trench[54]. The helicoidal burrow segments (Fig. 4E) in the here studied burrows would match an agrichnial behavior.

4. Vertical to horizontal, more than 15 cm-deep penetrating, thread-like burrows, ~0.5 mm in diameter are classified as *Trichichnus* isp. (Fig. 4C). They are locally present in Southern and Central Japan Trench. In CT-images, the commonly high-density of the burrow fill fits to the typical pyritization (or formation of other dense iron sulphide) of this trace[55], caused by intense localized microbial sulphate reduction, sustained by the originally organic-rich burrow lining[56]. *Trichichnus* has been interpreted as a burrow of a chemosymbiotic meiofaunal organism[57]. Alternatively, *Trichichnus* may represent pyrite-replaced long microbial sheaths[58].

5. Large, 4–17 cm-deep penetrating, vertical, straight to variably bending shafts, ca. 2–4 mm in diameter, are locally present in Southern Japan Trench. The burrows can show a single downward-oriented y-shaped branch (Supplementary Note 4).

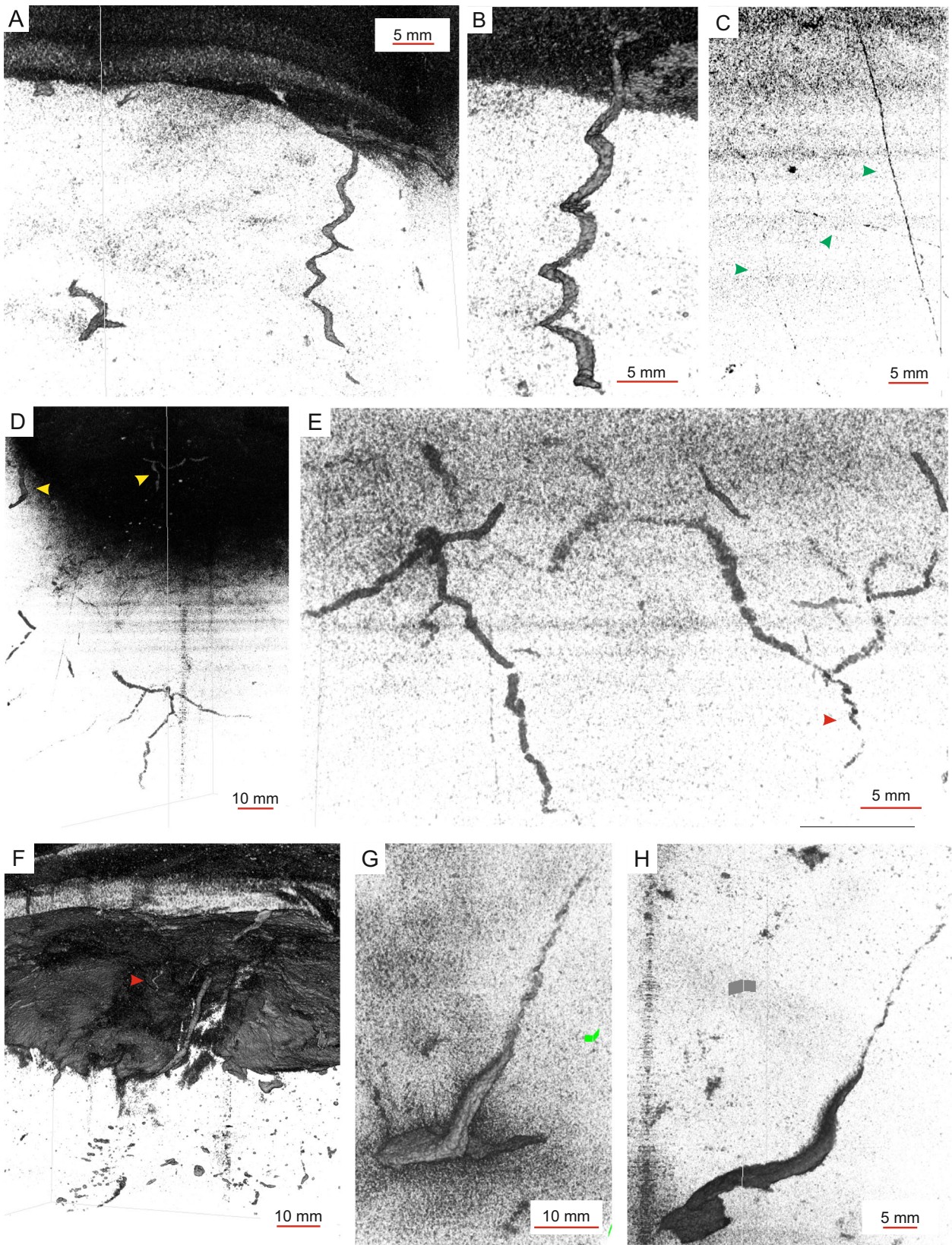

## Statistical relationships

Bioturbation properties (burrowing depth and trace density and diameter) and sedimentary C and N contents have the highest similarity at sites in Central Japan Trench according to non-metric multidimensional scaling (nMDS), but the similarity is lower in Southern and Northern Japan Trench (Fig. 5A). In addition, the bioturbation properties and sedimentary C and N contents are slightly more similar in expanded sections than condensed sections of the

**Fig. 4 | High-contrast 3D-views to traces assigned to the equilibrium suite.**
**A, B** Helicoidal burrows referred to as cf. *Gyrolithes locraensis*. Core section M0082D-1H-2WR7, 75.5–82.5 cm, Southern Japan Trench condensed section. **C** Long and narrow threads of *Trichichnus* isp. Core section M0089D-1H-5WR36, 34–40 cm, Central Japan Trench condensed section. **D, E** Irregular and variably oriented ramifying burrow system (*Pilichnus*), with local helicoidal segments (red arrow). Upwards, towards the colonization surface, the *Chondrites*-like traces can be associated with few mm wide vertical shafts (yellow arrow). Core section M0089D-1H-5WR35, 15–24 cm, Central Japan Trench condensed section. **F** A close up to the colonization surface illustrating vertical shafts related to *Pilichnus*. Red arrow points to a *Gyrolithes* fragment. The surface is affected by erosion generated by the subsequent gravity flow event. M0083F-1H-10WR16, 0–21 cm, Central Japan Trench expanded section. **G, H** A *Zoophycos*-like narrow lobate burrow. Core section M0082D-1H-3WR8, 18–24 cm, Southern Japan Trench condensed section. Note that the core margins have bent downwards due to coring process.

trench-basins (Fig. 5B). Sedimentary total C contents are highest in Southern Japan Trench (Fig. 5C), where also the diameter of *Artichnus* is largest (Fig. 5D). The other trace genera do not show statistically significant size differences between the trench basins. Also burrowing depth and trace density are not significantly different between the trench basins (Supplementary Note 5). Spearman rank correlation tests show no statistically significant relationships between bioturbation properties and sediment grain size and C, N and S contents (Fig. 5E). However, a weak negative correlation may exist between *Artichnus* diameter and sedimentary C content in the event bed level (Fig. 5F).

## Discussion

### Benthic colonization in the Japan Trench

The results show repetitive patterns in benthic colonization after gravity-flow deposition, which can be summarized in a characteristic sedimentological-ichnological succession as follows (Fig. 6):

1. Emplacement of an event bed by a slow-moving (lower flow regime), decelerating gravity flow that carried predominantly mixed silt-clay sized material (Fig. 6, start of phase 1a). Ripple cross-laminated silt and mud-floccule ripples indicate that the initial deposition typically took place by traction under dilute, expanding turbulent flows; water was entrained into a muddy gravity flow as it transversed the steep (<25°) slopes surrounding the hadal trench[26,59]. The early stage of gravity flows commonly experiences pulsed velocity fluctuations and sediment concentration. Such beds are predominantly unbioturbated, but sporadic deposit-feeding and escape structures may occur (Fig. 3A–C).

2. With further flow deceleration, the mud content of the flow increased damping or even suppressing turbulence as documented by low ripple amplitude and low foreset angles as well as gradation from cross-laminated to parallel-laminated and ungraded structureless mud[27]. Convolute lamination becomes common upward, being in line with elevated water content of the rapidly deposited beds. C/N ratios indicate that some laminated heterolithic interbeds contain material originating from the upper slope, but most of the sediment was derived from the lower trench slope[12,29–31]. The top of the interval is variably developed and can be represented by graded mud settled from suspension (Fig. 6, end of phase 1a). For instance, Oguri et al. (2013)[60] reported an up to 40 m-high suspension cloud 4 months after an earthquake-triggered gravity flow in the Japan Trench.

3. The gravity flow bed is first colonized by indistinct burrow mottling, indicating high sediment water content and that the early colonization took place during the waning stage of the gravity flow or immediately after event bed deposition (Fig. 6, phase 1b).

4. An early post-depositional, opportunistic colonization is commonly recorded by *Phycosiphon* isp. and/or *Nereites* isp. These traces were produced by mobile infaunal deposit feeders, which do not maintain an open connection to the seafloor and require oxygenated porewater (Fig. 6, phase 2a). Subsequent colonization is recorded particularly by Rosselichnid traces such as *Artichnus* (Fig. 6, phase 2b).

5. These early post-depositional burrows are cross-cut by traces of an equilibrium suite dominated by deeply penetrating, slender, straight and ramifying burrow systems (cf. *Gyrolithes lorcaensis*, *Pilichnus*, *Trichichnus*), interpreted to comprise chemosymbiotic

and agrichnial burrows, formed especially by small fauna, probably including thyasirid bivalves (Fig. 6, phase 3). The infills of the latter two burrows are inferred to consist of dense iron sulphides such as greigite or pyrite[56].

The general colonization pattern highlighted above is recurring throughout the Japan Trench: Phases 2 and 3 are present in 75% and 87% of the studied beds, in which the bed top is observable, respectively. There are no statistically significant differences in burrowing depth, trace density and diameter during Phase 2 between the trench basins (Supplementary Note 5), except *Artichnus* diameter being largest in Southern Japan Trench (Fig. 5D). In addition, nMDS similarity in bioturbation properties and sedimentary C and N contents is comparable between trench basins, although slightly higher in Central Trench Basin (Fig. 5A), and between expanded and condensed trench-basin sections (Fig. 5B). Overall, spatial differences in bioturbation are small and potentially result from locally enlarged sedimentary C contents as in the case of larger *Artichnus* size in Southern Japan Trench (Fig. 5C). Similarly, Spearman rank correlation tests show no statistically significant relationships between bioturbation properties and sediment grain size and C, N and S contents (Fig. 5E), apart from the possible weak negative correlation between *Artichnus* size and sedimentary C content in the event bed level (Fig. 5F). While more data are needed to fully understand the relationship between trace maker size and TOC content in the event bed level, the possible negative correlation could relate to stress factors such as increased population density and/or elevated oxygen consumption in the freshly-deposited C-rich sediment layers.

The main departure from the idealized colonization model is the lack of Phase 2 colonization in one quarter of the studied beds occurring locally in each trench segment. In nearly all beds where Phase 2 is lacking, the bed top contains structureless mud interval (F1C), typical of soup ground conditions. Therefore, substrate consistency is locally a key factor effecting opportunistic colonization. Nonetheless, these beds are occasionally colonized by Phase 3 equilibrium fauna (Fig. 4), pointing to subsequent substrate consolidation and low event frequency (e.g., every several years). The few cases of lacking Phase 3 are interpreted to reflect so high gravity-flow event frequency that the deposits remain unbioturbated or that only the early stages of colonization become buried and are now preserved as "frozen tiers" because the equilibrium suite could not develop.

We thus conclude that the colonization by endobenthos and their nutritional strategies are controlled mainly by event deposition and its frequency, and subsequent changes in substrate consistency, oxygenation and organic matter delivery. The change from the moderately diverse trace assemblage, characterized by deposit-feeding structures, to the deep-tier trace assemblage that displays microbe-dependent nutritional strategies is in line with the previously documented microbially mediated carbon degradation and the development of a strong redox stratification in the trench sediment column[8,61,62].

Several studies have demonstrated that gravity flows deliver considerable amounts of particulate organic carbon and fuel microbial activity in the hadal trenches[7,11,62]. The basin physiography, combined with the tectonically active, earthquake-prone setting, let hadal trenches receive considerable amounts of sediments since gravity flows easily originate at the adjacent mud-covered steep slopes and provide both an elevated input of marine organic matter

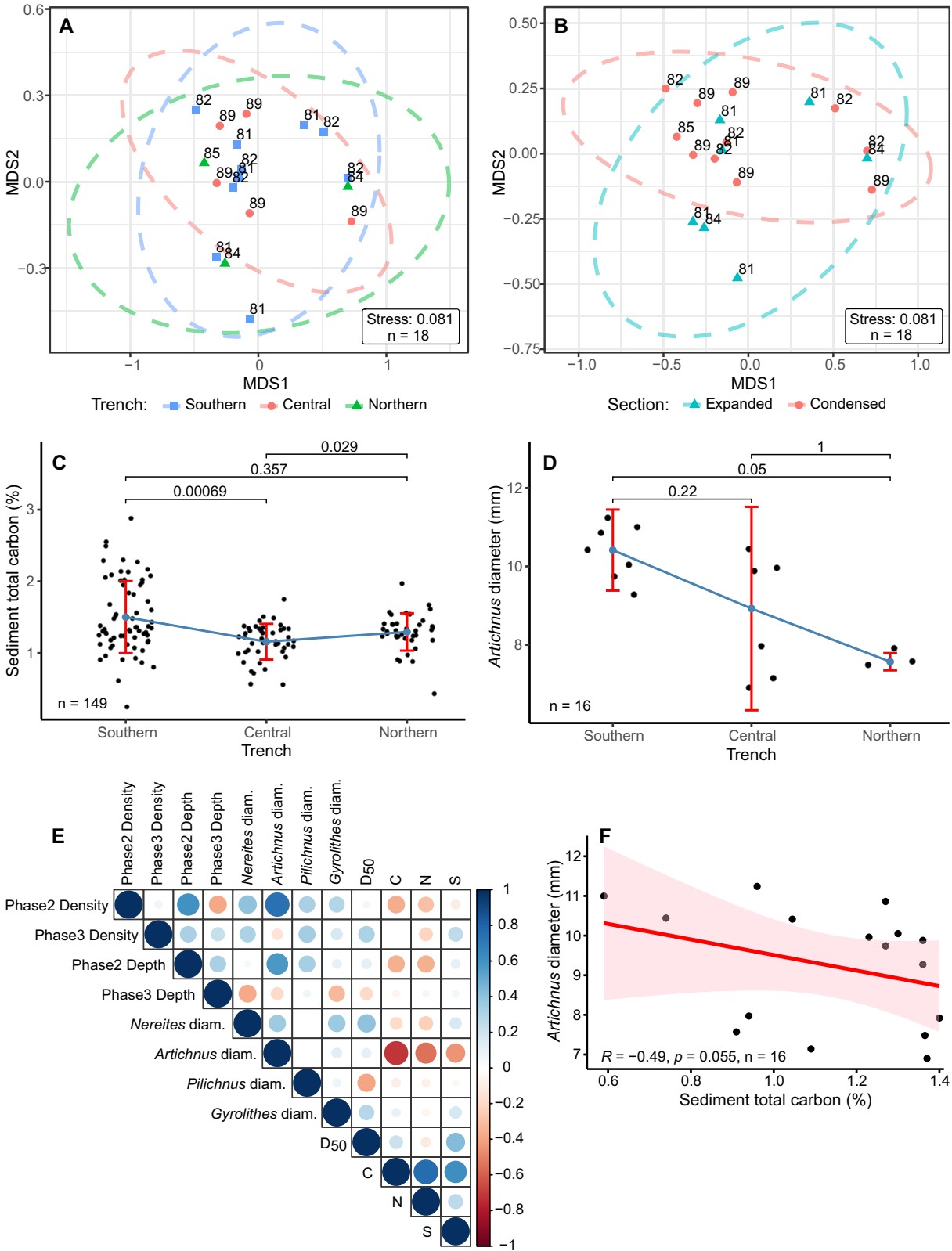

**Fig. 5 | Results of statistical analysis. A, B** Plots of non-metric multidimensional scaling results.The points are numbered according to coring sites. The closer the points plot to each other, the more similar they are. Ellipses indicate 95 % confidence intervals of the standard deviation. **C, D** Plots of sedimentary total C contents[28] and *Artichnus* diameter in Southern, Central and Northern Japan Trench. Red vertical bars indicate interquartile ranges. Median value points are connected by a blue line. The numbers are *p*-values of two-sided Wilcoxon rank sum tests with Bonferroni correction for multiple comparisons. **E** Spearman rank correlation matrix of bioturbation properties, sediment grain size median and total C, N and S contents. **F** Plot of sediment total C content versus *Artichnus* diameter with Spearman rank correlation coefficients and a linear regression line. The shaded band indicates the 95% confidence interval. *P*-values < 0.05 are considered statistically significant. n – number of observations. Source data for Fig. 5 are available as a Source Data file.

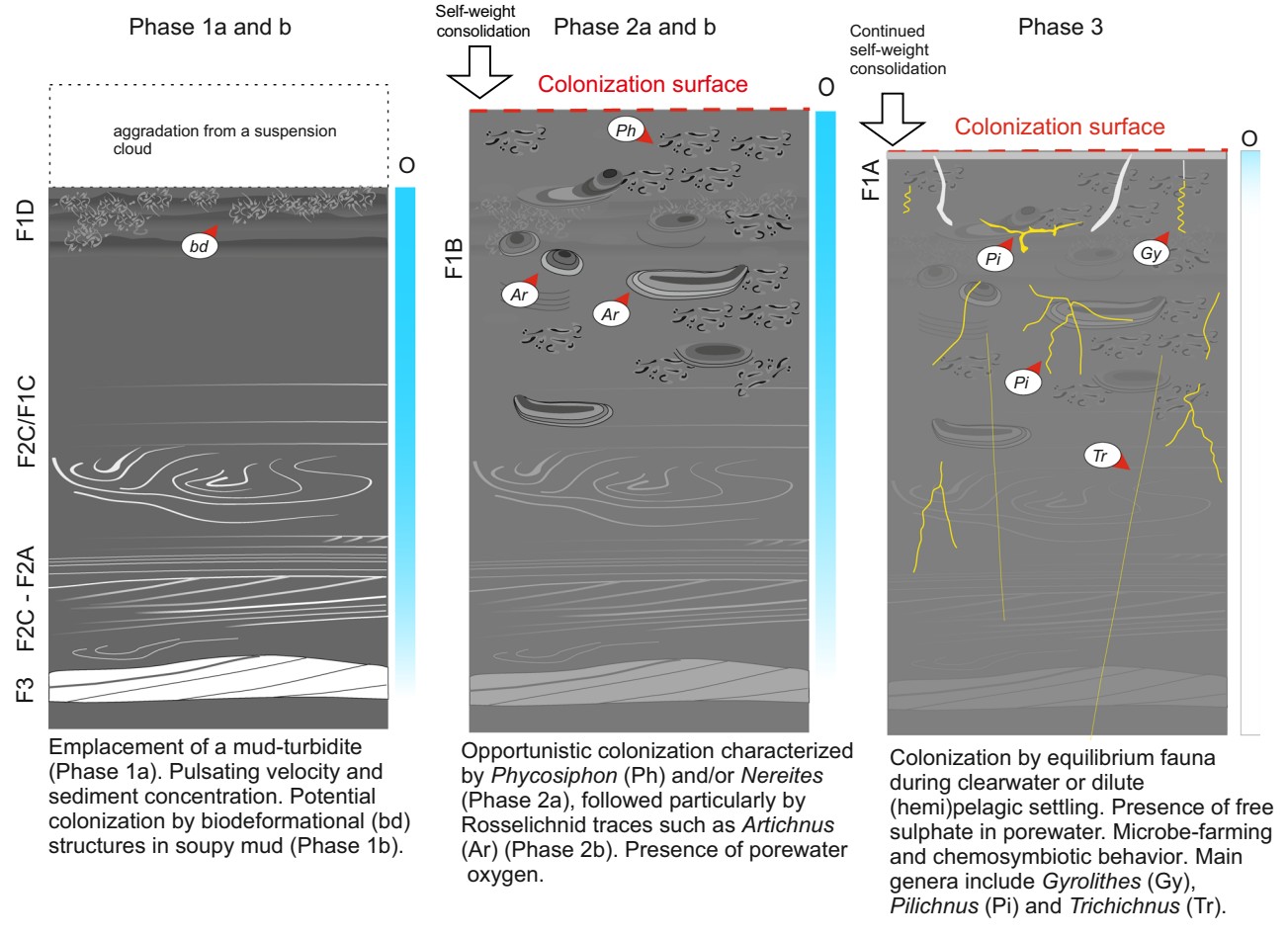

**Fig. 6 | A sedimentological-ichnological model for event bed colonization in the Japan Trench.** See text for further explanation. O–Oxygen.

as well as inorganic sediments[4]. Organic matter cycling and interlinked changes in redox conditions are considered as the main factors, shaping the microbial abundance and community composition in hadal trench deposits[8,9,61–64]. In the Japan Trench, porewater dissolved sulphate has been found to migrate downward several meters below the seafloor[7]. This fits to the observed variably-penetrating, now pyritized burrow systems (*Chondrites*-group traces, *Trichichnus*, cf. *Gyrolithes lorcaensis*), which were originally enriched in organic substances like faeces, mucus and extracellular polymeric substances, left behind by the burrowing organisms, and microbial communities along the burrow walls[52,65,66]. These organic substances sustained localized microbial reduction of the porewater sulphate, leading to hydrogen sulphide supersaturation and iron-sulphide precipitation in the burrows[56,67]. The localized iron-sulphide precipitation started within the top few meters of the sediment column before the burrows were buried below the zone of free dissolved sulphate in the porewater[56]. In addition, high methane concentrations have been found below the zone containing dissolved sulphate[7] (See Supplementary Note 1), which would support additional iron-sulphide overgrowth on burrows by anaerobic methane oxidation at the sulphate-methane transition zone[68].

For the sake of an analogy, gravity-flow deposition thus proves to have a similar impact on hadal trench benthic habitats as forest fires in the terrestrial realm; the fires reset vegetation successions and change key ecological parameters such as light, temperature and nutrient availability. In the hadal environment, the gravity flows regulate oxygen and organic matter delivery and control the substrate consistency. In response to the changing ecological parameters through time, the endobenthic ethological strategies grade from initial opportunistic,

infaunal deposit feeding in oxic sediments to chemosymbiont-supported nutritional strategies.

## Differences to the Abyssal zone

Considering the contrasting basin physiography and corresponding differences in organic matter delivery, sedimentation rate, microbial communities, early diagenesis and substrate consistency, it is not surprising that bioturbation and colonization successions differ in many respects between the hadal and abyssal zones. Similarities, however, exist in terms of the traces present at ichnogenus level: *Phycosiphon*, *Nereites*, *Artichnus*, *Pilichnus*, *Trichichnus*, *Zoophycos* and *?Phycodes* are all reported from the "deep sea" or abyssal sediments[57,69].

While the holothurian trace *Artichnus* have been reported from some "deep sea" trace fossil assemblages, their dominance in bioturbational fabrics is not as typical for the abyssal zone. In the studied samples, *Artichnus* and other Rosselichnid burrows are the largest traces and account for highest degree of biogenic reworking throughout Japan Trench. They are present in approximately 70% of the studied top preserved beds and invariably result in 100% trace density and on average more than 20 cm thick bioturbated zones. The difference is likely due to the above-mentioned elevated accumulation of organic matter, owing to the trench basin morphology, which reportedly increases the number of deposit feeders on the trench floor in comparison to the adjacent abyssal plains[4,70]. In the hadal zone, this is particularly in favour of the holothurians, which dominate the macrofaunal biomass, especially in the deepest trenches[1]. Considering the commonness of *Artichnus*, the burrowing holothurians have the potential of being ecosystem engineers, impacting nutrient cycling in the hadal zone.

Other contrasting ichnotaxonomic aspects include the helicoidal burrow morphology that is recurring in the studied hadal deposits, but rarely reported from abyssal plains. The type material of *Gyrolithes* was recovered from Cretaceous pelagic, open-marine chalk in Belgium and additional specimens are known from other chalk deposits[71]. Recently, *Gyrolithes* has been described for instance from the Miocene turbiditic Marnoso-arenacea Formation in Italy[72,73], where it is associated with methane-seep deposits. Furthermore, Seilacher (2007)[74] interpreted the draft fill of some *Gyrolithes* as being indicative of a farming behavior of animals being capable to utilize the steep geochemical (redox) gradient between oxygenated water in the *Gyrolithes* tube and the reducing host sediment. Its recurring occurrence in the hadal zone sediments, particularly together with *Pilichnus and Trichichnus*, is also thought to reflect the elevated endobenthic microbial activity in the hadal trenches, which bolster endobenthic agrichnial behaviour. In contrast, pre-depositional burrows in abyssal deposits are characterized by complex near-surface burrows (graphoglyptids) interpreted to represent agrichnial behavior as an adaptation to nutrition-scarce conditions[69,75,76]. Graphoglyptids were not observed in the studied CT-images and all agrichnial and chemichnial burrows appear to be related to contrasting microbial abundances in sediment. The higher amount of organic carbon increases benthic oxygen uptake in the hadal trenches relative to the surrounding abyssal areas[61]. This leads to low oxygen penetration depth during the late-stage colonization, explaining the varied burrowing depths of pre-depositional burrows in the Hadal zone.

Similarly, *Rhizocorallium* does not commonly occur in deep-marine settings, even in gravity-flow deposits. Nonetheless, it has been reported from different Cenozoic flysch successions in Spain, Switzerland, Poland and Turkey[77–80] (see Uchman, 2004 for further occurrences). Modern equivalents have been found in the deep South China Sea[81]. It is unclear whether the gravity-flow modulated hadal environment is conducive to *Rhizocorallium* producers, or whether the single trace was made by a rare individual that was imported to the site from shallower depths by the gravity-flow.

To conclude, the studied record of bioturbational structures in hadal trenches reveals intensely burrowed fabrics and recurring colonization patterns. Gravity flow deposition and its frequency, and consequent changes in substrate consistency, oxygenation, organic matter delivery and its degradation controlled the subsequent colonization by benthic fauna in the Japan Trench. The mud-rich gravity flow deposits are colonized by repeating burrow successions: the initial colonization is characterized by biodeformational structures and/or feeding structures like *Phycosiphon* and *Nereites*, followed particularly by Rosselichnid traces such as *Artichnus*. Slender helicoidal, lobate and deep-penetrating straight and ramifying burrow systems like cf. *Gyrolithes lorcaensis*, *Pilichnus* and *Trichichnus* represent burrows of microbe-farming and chemosymbiotic behavior of invertebrates, exploiting the sulfate reduction zone; they document the final colonization stage. Local departures from this model are caused by soupy substrate and high event frequency, which may prevent colonization. The endobenthic ethological strategies thus grade from initial opportunistic, infaunal deposit feeding in oxic sediments to chemosymbiont-supported nutritional strategies, likely driven by organic matter degradation and post-event upward expansion of the anoxic zone. The resulting bioturbational fabric shows commonly decimeter-scale completely bioturbated intervals. In particular, burrowing holothurians producing *Artichnus* mix typically 20 cm-thick sediment zones and may significantly impact nutrient cycling in hadal trenches. Potential chemosymbionts include thyasirid bivalves.

Up to now, hadal environment has been the only bathymetric zone of which we have not had any ichnological data. Our data show that ichnological models of the abyssal plains cannot be simply extrapolated to hadal trenches. In future, additional ichnological investigations of hadal sediments are needed to distinguish trench-specific factors from characteristics common to all hadal trench environments. In particular, more data are needed from trenches characterized by oligotrophic conditions as well as those showing variable substrate lithology. A coherent understanding of bioturbation is essential to fully comprehend habitat-modifying biotic-abiotic processes and ecosystem functioning in the Hadal zone.

## Methods

### Research materials

The Japan Trench is located at the plate boundary where the Pacific Plate is subducting beneath the Okhotsk Plate with a convergence rate of 8.0–8.6 cm/yr[30]. It is 611 km long and has a surface area of 37854 km² [6]. Normal fault grabens resulting from the flexural bending of the slightly obliquely subducting Pacific Plate oceanic crust form numerous isolated basins that act as terminal sinks for trench-fill sedimentation along the trench axis of the Japan Trench[12]. Sediment cores were collected with up to 40 m long giant piston corers on R/V Kaimei during the International Ocean Discovery Program (IODP) Expedition 386 Japan Trench Paleoseismology in 2021 (Fig. 1)[24].

### Approach

This study is primarily based on sedimentological and ichnological documentation of micro-CT -imaged, 20 whole-round core sections (each 17–55 cm long). The selected core intervals were retrieved from (1) relatively expanded sections within the main depocenters (experiencing the highest sedimentation rates) and from (2) more condensed sections in a relative bathymetric high in the trench-basins of Southern, Central and Northern Japan Trench (SI 1). The data are supported with grain size analyses and C/N data. Moreover, the data are integrated with previously published core logs, multibeam bathymetric images and seismic sub-bottom profiles[24] to place the studied core sections in sedimentologic and stratigraphic context (Supplementary Note 1).

### Sedimentology and Ichnology

Sedimentological approach included descriptions of the inferred lithology, grain size (visual estimation, measurements; see below) and its trend, primary and secondary sedimentary structures and bedding contacts. The described sedimentary facies are presented in Supplementary Note 2.

Ichnological data collection included description of bioturbational structures (incipient ichnofossils), ichnogenera, cross-cutting relationships, suites, bioturbation index (BI), and trace density. The BI classification scheme[33] includes the following classes: BI 0 (0% reworked); BI 1 (1–4% reworked); BI 2 (5–30% reworked); BI 3 (31–60% reworked); BI 4 (61–90% reworked; BI 5 (91–99%) and BI 6 (100% reworked). Trace densities of the colonization phases 2 and 3 were quantified by counting the occurrence of traces in a 10 × 10 square grid[82] in the center cross-section of a core. The described trace morphologies are presented in Supplementary Note 4.

### Computed tomography imaging

CT-images of whole-round core sections were taken using a GE phoenix v|tome|x s. The samples were imaged using a 240 kV microfocus tube with an accelerating voltage of 200 kV and a tube current of 1 mA for a total power of 200 W. 0.5 mm of copper and 0.5 mm of aluminum were used as a beam filter. Some of the samples were so long that they had to be scanned twice and rotated 180 degrees between scans, with single scans consisting of one or two vertically displaced scans for a total amount of 1–4 scans per sample, depending on sample length. For all the single scans, 2500 projections were taken during a 360-degree rotation and at each angle the detector waited for a single exposure time and then took an average over three exposures. With a single exposure time of 200 ms this resulted in a total time per scan of 33 min. The obtained spatial resolution was 100 μm. The beam

hardening correction coefficient used in reconstruction was 8 and no ring-artefact reduction was required. The resulting images were analyzed and inspected with ThermoFisher PerGeos 2020.2 software.

### C, N and grain size analysis

The sampling targeted identified sedimentary facies, and the samples were sent for analyses to the Eurofins Labtium, Finland. Contents of C and N in the sediment samples were measured by high-temperature combustion analysis using an Elementar vario MAX cube analyzer, following the ISO 16948 method. Sediment grain size distribution was determined down to 0.6 μm using a Micromeritics Sedigraph III 5120 X-ray absorption sedimentation analyzer. The samples were freeze-dried and pretreated with excess hydrogen peroxide to remove organic matter prior to the analysis. No sieving was performed because of the fine grain size of the samples. Median grain size ($D_{50}$) and geometric sorting ($\sigma_G$) were derived using the GRADISTAT software[83].

### Statistical analysis

Similarities of Phase 2 and 3 burrowing depths and trace densities, the diameters of *Nereites*, *Artichnus*, *Pilichnus* and *Gyrolithes*, and sedimentary C and N contents among groups were assessed by non-metric multidimensional scaling (Bray-Curtis dissimilarity, two dimensions, 200 iterations) using the package `vegan` 2.6-8 in R 4.1.3[84]. Relationships between bioturbation properties and sediment grain size and total C, N and S contents were explored using Spearman rank correlation diagrams, and the relationships were detailed using Kruskal-Wallis rank sum tests and pairwise Wilcoxon rank sum tests with Bonferroni correction for multiple comparisons.

### Reporting summary

Further information on research design is available in the Nature Portfolio Reporting Summary linked to this article.

## Data availability

The main data set is presented in the main article and as supplementary items. The supporting C, N and grain size data are archived in Pangaea[28] (https://doi.org/10.1594/PANGAEA.967716). The cores are stored in the IODP core repository in Japan (https://www.kochi-core.jp/en/iodp-curation/index.html). Source data for Fig. 5 are available as a Source Data file. All IODP mission-specific platform data are accessible at http://iodp.pangea.de. CT-images of the whole-round core sections are available at the Fairdata.fi portal: https://doi.org/10.23729/22b512b4-58c0-447d-849d-360c6d645c22. Source data are provided with this paper.

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

## Acknowledgements

This research used samples and data provided by the International Ocean Discovery Program (IODP). CT-images of the whole-round sections were provided by the Geological Survey of Finland (G.T.K.). We thank Jukka Kuva for CT-scanning. X.C.T. was funded by the Research Council of Finland via the RAMI infrastructure project (#293109). This research was funded in part by the Austrian Science Fund (F.W.F.) [grant https://doi.org/10.55776/P36809] to MS and MM and by Japanese funding source JSPS KAKENHI JP23K22586 to K.I.

## Author contributions

J.H., J.V. and A.W. described and interpreted the data and wrote the paper with contributions from M.M., M.S., J.N.P., and K.I.; M.M. and M.S. provided site summary data. J.V. curated the WR-samples and acted as the study manager.

## Competing interests

The authors declare no competing interests.
