## [Peer Review file · Nature Communications]

Bioturbation in the hadal zone

Corresponding Author: Dr Jussi Hovikoski

Version 0:

Reviewer comments:

Reviewer #1

(Remarks to the Author)

General comments

The manuscript presents, for the first time, the record of bioturbation in the hadal zone. It is, without a doubt, the most innovative work I have seen in ichnology in recent years and must be published. The work is well written, the methodology well explained, the images of excellent quality (including supplementary material), and the characterizations of the ichnofossils very well done, leaving no doubt as to the composition of the recorded ichnofauna in the Japan Trench. The discussion is clearly supported by the data. Only two main aspects should be addressed before final acceptance:

1. It is impressive, in the record, the occurrence of patterns of excavations classically found in shallower areas, such as Gyrolithes, Rhizocorallium and rosselid burrows, which normally not occur in deep-sea ichnocoenoses. However, nothing is said about it in the discussion. It would be very important to know how the authors explain this occurrence in the discussion, to create a basis for the revision of the paradigm that these ichnogenera would be unique to shallow marine environments.

2. The 'Conclusions' are resembling 'Final remarks'. There are several issues in the conclusion that are rather discussion, mostly those presented in lines 343-344 and 356-358. I recommend making the conclusions more concise and move all complementary information to the discussion.

Specific comments/corrections needed

1. Text

Line 432 – italics in Chondrites

Lines 440, 451, 518, 530, 604 – italics in Zoophycos

Line 442 – Bromley, 1990 – not cited in the text.

Line 449 – italics in Thyasira

Lines 499-502 – Jumars et al., 1990 – not cited in the text.

Line 504 – italics in Trichichnus

Lines 505-512 – review the alphabetical order of the references.

Lines 520, 536 – italics in Gyrolithes

Line 550 – italics in Nereis diversicolor (separate both words) and N. virens

Line 595 – italics in Nereites

Line 599 – italics in Tisooa siphonalis

Line 615 – italics in Artichnus pholeoides

Line 643 – review the use of "&"; "and" has been used throughout the whole text

The following cited references are missing:

Bromley, 1996

Meadows et al., 2012

Mikuláš et al., 2004

Nierop et al., 2017

Uchman, 1999

Xu and Fang, 2018

2. Supplementary material:

Table SI2A – put all ichnotaxa names in italics. Review the use of “&” in some citations

Fig. SI2B-O – *Trichichnus* – correct the spelling.

Reviewer #2

(Remarks to the Author)

This paper reads easily and packs a large volume of observations throughout different hadal basins. Also, it offers some clues about how benthic fauna and sediment interaction occurs in this underexplored areas. As the authors stated correctly in the introduction there is a lack of documentation of modern bioturbation in the hadal zone. Thus, this manuscript offers new and first (at least to my knowledge) detailed ichnological data on hadal environments. I like the sedimentological-ichnological model for event bed colonization that the authors propose and I agree with most of the traces described. The authors discuss and explain with good arguments the importance of gravity flows events for explained trace fossil distributions. Having said that, I cannot recommend this manuscript unless major changes are made in relation to reporting the data in a more quantitative way and proving the results with following statistical analysis since the authors report variability between and within the three basins (see below).

My main concerns are in relation to the methodology and the subsequent presentation of the results and their discussion. The authors base their results on the description of the traces (and their correlation with facies features both sedimentological and geochemical). Then, the authors conduct the discussion. However more quantitative data and statistical analysis (see below) are needed to go beyond the model presented so that the environmental discussion does not fall short, especially if the idea is to take it to a regional or more global perspective (i.e., application to other hadal zones). One of the main discoveries of the authors is slightly exposed in the paragraph 306 and yet maybe the most important “The ichnological colonization successions show variability both in successive gravity flows and between trench-basins.....”. A variability is observed between hadal settings even the sedimentological-ichnological model for event bed colonization seems to be the same. Thus, this variability needs to be characterize. Especially considering the outstanding dataset that the authors have. In other words, additional statistical evidence is needed to expose the variability of this model and the factors behind.

For example, the authors say that “The between-facies differences are small, but the highest TOC contents and C/N ratios were measured in F2”. But I wonder what about variations in TOC content within the same facies at the different locations and if these have a significant correlation with trace fossil features (e.g., density, size). Same for other features that the authors have data (e.g., grain size, C/N). I recommend plotting the TOC and C/N in the Supplementary Material logs. So my suggestion would be to characterize density and size of the different ichnogenera (in the three phases exposed in Fig. 5 and along the three basins). Then for example, a Spearman rank correlation can be conducted to check the correlation of TOC, grain size, C/N with the different ichnogenera features. Also, using a density approach will allow to conduct a non-metric multidimensional scaling (n-MDS) between locations. Proving the similarity or dissimilarity between basins or within the same basin between trench axis and condensed sections.

Other comments:

In the Line 60 the authors indicate that they are going to study the ichnofabrics. Then throughout the manuscript (results and discussion) this approach is never mentioned again. Finally, in the Line 340 (Conclusions), the authors talk about ichnofabrics and in the paper there have not been defined. I suggest modified this part and also the introduction. This manuscript does not conduct an ichnofabric approach.

The conclusions are too long in my opinion (even longer than some sections of the discussion). For example, paragraphs like 347 could be removed or modified to be more concise (by removing some writing flourishes). Also other paragraphs (e.g., 355) should be placed in the discussion.

Minor

Lines 54-57: this sentence is confusing. “it can be crucial for the existence of other organisms” What other organisms? You mention echinoids and holothurians but what type of organisms are they influencing?

Lines 328-329: Delete this sentence. The abyss is not flat. Mapping surveys of the seafloor have revealed the huge heterogeneity of the deep sea (Riehl et al., 2020). Recent estimates suggest that globally, there are around 25 million hill/mountain-scale seafloor features taller than 100 m (Wessel et al., 2010), that may extend over 41% of the ocean floor Harris (2014).

About Zoophycos: If the spreiten cannot be clearly distinguishable, I would suggest Zoophycos-like. In the video looks to me more like a small Spirophyton. I leave it to the author’s consideration.

I hope to read a new version of this interesting manuscript

Version 1:

Reviewer comments:

Reviewer #1

(Remarks to the Author)

As far as I could determine, the authors implemented all of the reviewers' suggestions to improve the text. The discussion about the occurrence of certain ichnotaxa found in shallow habitats in the hadal zone is satisfactory. The conclusions were trimmed to be more objective, which benefited the paper. In my opinion, it is suitable for publication, except for a correction still needed: in line 368, correct the spelling of *Trichichnus* (not *Tricichnus*).

Reviewer #2

(Remarks to the Author)

I read with this new version of the manuscript and I value the effort made by the reviewers. I see that they have included an appropriate statistical treatment to give robustness to their data. In the new statistical methodology the authors use correctly n-MDS for bioturbation features and C/N ratios.

However, then grain size and TOC content correlations are missing. This would not be important if it were not for the results shown in Fig. 5C. Following these results (Fig. 5C) and considering that you have significant differences in TOC content between the central trench and the other ones (P-values <0.05), why don't explore the correlation between TOC and your bioturbation properties (burrowing depth and trace density and diameter).

Fig. 5C only shows the variation of one environmental factor (i.e., TOC) across the three trenches, it does not show anything about bioturbation. I suggest modifying it and adding something that shows the correlation (if any) between bioturbation features and grain size and TOC as I mentioned in my previous review. Even if you don't have any correlation between TOC and bioturbation features it is important to show it. Then for example line 263 "Overall, spatial differences in bioturbation are small and potentially result from locally enlarged TOC contents as in the case of larger *Artichnus* size in Southern Japan Trench (Fig. 5C)." would not look disconnected since Fig. 5C does not show small bioturbation differences.

You mention in Line 676 Spearman rank correlation diagrams but I do not see them in the manuscript. Maybe I missed them. With the Spearman rank correlation you can see the correlation between TOC and bioturbation properties and also grain size (which is missing from the manuscript). Then, you can add some weight to your discussion (Paragraph 257).

In short, perhaps I am confused and have not found it, but otherwise I suggest that the authors explore how TOC variations (which exist in the trenches as they showed in Fig. 5C) correlate with bioturbation features. This is truly important since the authors claim that organic matter delivery is a key factor during endobenthic colonization. Also explore the correlation with grain size (data they obtained for this study). The authors mention in Line 220 that "The other trace genera do not show statistically significant size differences between the trench basins. Also burrowing depth and trace density are not significantly different between the trench basins.". But are the size, burrowing depth, density variations correlated with TOC or grain size?

Overall I like the manuscript and the benthic colonization model that the authors propose for hadal environments.

Minor

In the line 220 the authors mention that "Sedimentary C and N contents are highest in Southern Japan Trench (Fig. 5C)" but figure 5C represents Total carbon. Please rephrase.

Fig. 5D *Artichnus* should be in italics.

Version 2:

Reviewer comments:

Reviewer #2

(Remarks to the Author)

I am pleased to read this latest version of the manuscript. I believe that the discussion is clearly supported by the data now. Thus, I think it is suitable for publication. Well done.

Minor.

Be careful in Fig. 5.

Ichnotaxa should be in italics in some of the legend plots. For example, Fig. 5D and F *Artichnus* should be in italics.

Same for Fig. 5F ichnotaxa.

Fig. 5 F should be on the left bottom corner considering alphabetical order. Or changing the order of the plots; now it seems a bit confusing.

RESPONSE TO REFEREES

Our replies to the reviewer comments are written in red in the text below.

REVIEWER COMMENTS

Reviewer #1 (Remarks to the Author):

General comments

The manuscript presents, for the first time, the record of bioturbation in the hadal zone. It is, without a doubt, the most innovative work I have seen in ichnology in recent years and must be published. The work is well written, the methodology well explained, the images of excellent quality (including supplementary material), and the characterizations of the ichnofossils very well done, leaving no doubt as to the composition of the recorded ichnofauna in the Japan Trench. The discussion is clearly supported by the data. Only two main aspects should be addressed before final acceptance:

1. It is impressive, in the record, the occurrence of patterns of excavations classically found in shallower areas, such as *Gyrolithes*, *Rhizocorallium* and rosselid burrows, which normally not occur in deep-sea ichnocoenoses. However, nothing is said about it in the discussion. It would be very important to know how the authors explain this occurrence in the discussion, to create a basis for the revision of the paradigm that these ichnogenera would be unique to shallow marine environments.

Thank you very much for your constructive comments. We have now added a discussion of *Gyrolithes*, *Rhizocorallium* and rosselid burrows in deep sea (see section Differences to the Abyssal zone).

2. The 'Conclusions' are resembling 'Final remarks'. There are several issues in the conclusion that are rather discussion, mostly those presented in lines 343-344 and 356-358. I recommend making the conclusions more concise and move all complementary information to the discussion.

The Conclusions sections is now shortened and partly reorganized, as suggested by both reviewers.

Specific comments/corrections needed

1. Text

Line 432 – italics in Chondrites

Lines 440, 451, 518, 530, 604 – italics in Zoophycos

Line 442 – Bromley, 1990 – not cited in the text.

Line 449 – italics in Thyasira

Lines 499-502 – Jumars et al., 1990 – not cited in the text.
Line 504 – italics in *Trichichnus*
Lines 505-512 – review the alphabetical order of the references.
Lines 520, 536 – italics in *Gyrolithes*
Line 550 – italics in *Nereis diversicolor* (separate both words) and *N. virens*
Line 595 – italics in *Nereites*
Line 599 – italics in *Tisooa siphonalis*
Line 615 – italics in *Artichnus pholeoides*
Line 643 – review the use of “&”; “and” has been used throughout the whole text

These are corrected.

The following cited references are missing:

Bromley, 1996
Meadows et al., 2012
Mikuláš et al., 2004
Nierop et al., 2017
Uchman, 1999
Xu and Fang, 2018

These are corrected.

2. Supplementary material:

Table SI2A – put all ichnotaxa names in italics. Review the use of “&” in some citations
Fig. SI2B-O – *Trichichnus* – correct the spelling.

These are now corrected.

Thank you!

Reviewer #2 (Remarks to the Author):

This paper reads easily and packs a large volume of observations throughout different hadal basins. Also, it offers some clues about how benthic fauna and sediment interaction occurs in this underexplored areas. As the authors stated correctly in the introduction there is a lack of documentation of modern bioturbation in the hadal zone. Thus, this manuscript offers new and first (at least to my knowledge) detailed ichnological data on hadal environments. I like the sedimentological-ichnological model for event bed colonization that the authors propose and I agree with most of the traces described. The authors discuss and explain with good arguments the importance of gravity flows events for explained trace fossil distributions. Having said that, I cannot recommend this manuscript

unless major changes are made in relation to reporting the data in a more quantitative way and proving the results with following statistical analysis since the authors report variability between and within the three basins (see below).

My main concerns are in relation to the methodology and the subsequent presentation of the results and their discussion. The authors base their results on the description of the traces (and their correlation with facies features both sedimentological and geochemical). Then, the authors conduct the discussion. However more quantitative data and statistical analysis (see below) are needed to go beyond the model presented so that the environmental discussion does not fall short, especially if the idea is to take it to a regional or more global perspective (i.e., application to other hadal zones).

One of the main discoveries of the authors is slightly exposed in the paragraph 306 and yet maybe the most important “The ichnological colonization successions show variability both in successive gravity flows and between trench-basins.....”. A variability is observed between hadal settings even the sedimentological-ichnological model for event bed colonization seems to be the same. Thus, this variability needs to be characterize. Especially considering the outstanding dataset that the authors have. In other words, additional statistical evidence is needed to expose the variability of this model and the factors behind.

For example, the authors say that “The between-facies differences are small, but the highest TOC contents and C/N ratios were measured in F2”. But I wonder what about variations in TOC content within the same facies at the different locations and if these have a significant correlation with trace fossil features (e.g., density, size). Same for other features that the authors have data (e.g., grain size, C/N). I recommend plotting the TOC and C/N in the Supplementary Material logs.

So my suggestion would be to characterize density and size of the different ichnogenera (in the three phases exposed in Fig. 5 and along the three basins). Then for example, a Spearman rank correlation can be conducted to check the correlation of TOC, grain size, C/N with the different ichnogenera features. Also, using a density approach will allow to conduct a non-metric multidimensional scaling (n-MDS) between locations. Proving the similarity or dissimilarity between basins or within the same basin between trench axis and condensed sections.

Thank you very much for very good suggestions. We have now conducted statistical analyses (see the new method section) and modified the discussion section.

We carried out non-parametric correlation analyses and multiple comparisons (Wilcoxon rank sum tests with Bonferroni correction) to test statistical relationships between studied variables.

We ran non-metric multidimensional scaling (n-MDS) analysis to explore the similarity of trace density, burrowing depth, diameter of key taxa, and sedimentary total C, N and S contents in each top preserved bed at the studied trench basins. In both Phase 2 and

Phase 3, the burrowing depth commonly exceeds the length of the studied micro-CT-scanned samples. For this reason, the burrowing depth of Phase 2 traces was sometimes estimated by combining the micro-CT images with adjacent lower-resolution CT images. Phase 3 traces are only visible in micro-CT images and for those the lower-resolution CT images could not be used and the number of observations is slightly lower (see Source Data.xlsx for actual data).

The results are shown in the new figure 5 and discussed in the Discussion. TOC and C/N are also added to the Supplementary logs.

Other comments:

In the Line 60 the authors indicate that they are going to study the ichnofabrics. Then throughout the manuscript (results and discussion) this approach is never mentioned again. Finally, in the Line 340 (Conclusions), the authors talk about ichnofabrics and in the paper there have not been defined. I suggest modified this part and also the introduction. This manuscript does not conduct an ichnofabric approach.

We used the word "ichnofabric" as a general term for bioturbational fabric. We did not mean ichnofabric analysis. The word ichnofabric is now removed for clarity.

The conclusions are too long in my opinion (even longer than some sections of the discussion). For example, paragraphs like 347 could be removed or modified to be more concise (by removing some writing flourishes). Also other paragraphs (e.g., 355) should be placed in the discussion.

The Conclusions section is now shortened and partly reorganized, as suggested by both reviewers.

Minor

Lines 54-57: this sentence is confusing. "it can be crucial for the existence of other organisms" What other organisms? You mention echinoids and holothurians but what type of organisms are they influencing?

The sentence is now removed.

Lines 328-329: Delete this sentence. The abyss is not flat. Mapping surveys of the seafloor have revealed the huge heterogeneity of the deep sea (Riehl et al., 2020). Recent estimates suggest that globally, there are around 25 million hill/mountain-scale seafloor features taller than 100 m (Wessel et al., 2010), that may extend over 41% of the ocean floor Harris (2014).

Ok, good point, the sentence is deleted.

About Zoophycos: If the spreiten cannot be clearly distinguishable, I would suggest Zoophycos-like. In the video looks to me more like a small Spirophyton. I leave it to the

author's consideration.

Ok, we use Zoophycos-like.

I hope to read a new version of this interesting manuscript

Thank you!

REVIEWER COMMENTS

Reviewer #1 (Remarks to the Author):

As far as I could determine, the authors implemented all of the reviewers' suggestions to improve the text. The discussion about the occurrence of certain ichnotaxa found in shallow habitats in the hadal zone is satisfactory. The conclusions were trimmed to be more objective, which benefited the paper. In my opinion, it is suitable for publication, except for a correction still needed: in line 368, correct the spelling of Trichichnus (not Tricichnus).

Thank you!

Reviewer #2 (Remarks to the Author):

I read with this new version of the manuscript and I value the effort made by the reviewers. I see that they have included an appropriate statistical treatment to give robustness to their data. In the new statistical methodology the authors use correctly n-MDS for bioturbation features and C/N ratios.

However, then grain size and TOC content correlations are missing. This would not be important if it were not for the results shown in Fig. 5C. Following these results (Fig. 5C) and considering that you have significant differences in TOC content between the central trench and the other ones (P-values <0.05), why don't explore the correlation between TOC and your bioturbation properties (burrowing depth and trace density and diameter). Fig. 5C only shows the variation of one environmental factor (i.e., TOC) across the three trenches, it does not show anything about bioturbation. I suggest modifying it and adding something that shows the correlation (if any) between bioturbation features and grain size and TOC as I mentioned in my previous review. Even if you don't have any correlation between TOC and bioturbation features it is important to show it. Then for example line 263 "Overall, spatial differences in bioturbation are small and potentially result from locally enlarged TOC contents as in the case of larger Artichnus size in Southern Japan Trench (Fig. 5C)." would not look disconnected since Fig. 5C does not show small bioturbation differences.

You mention in Line 676 Spearman rank correlation diagrams but I do not see them in the manuscript. Maybe I missed them. With the Spearman rank correlation you can see the correlation between TOC and bioturbation properties and also grain size (which is missing from the manuscript). Then, you can add some weight to your discussion (Paragraph 257). In short, perhaps I am confused and have not found it, but otherwise I suggest that the authors explore how TOC variations (which exist in the trenches as they showed in Fig. 5C) correlate with bioturbation features. This is truly important since the authors claim that organic matter delivery is a key factor during endobenthic colonization. Also explore the

correlation with grain size (data they obtained for this study). The authors mention in Line 220 that “The other trace genera do not show statistically significant size differences between the trench basins. Also burrowing depth and trace density are not significantly different between the trench basins.”. But are the size, burrowing depth, density variations correlated with TOC or grain size?

We have added a Spearman rank correlation matrix to Figure 5 (new Figure 5E), which summarizes statistical relationships between bioturbation properties and sediment grain size and C, N and S contents. The grain size data have been added to the updated Source data file.

The bioturbation properties do not seem to be correlated with sediment properties in our dataset (except *Artichnus* and total C; Fig. 5) as we discuss in the text. We added plots with statistical comparisons of bioturbation properties in the trench basins in the electronic supplement (Supplementary Item 5) although there were no statistically significant differences (except *Artichnus* diameter in Fig. 5D).

A weak negative correlation between *Artichnus* diameter and C content may exist in the event bed level ($P=0.055$). This possible correlation is shown in the new Figure 5F and mentioned in the discussion.

Overall I like the manuscript and the benthic colonization model that the authors propose for hadal environments.

Minor

In the line 220 the authors mention that “Sedimentary C and N contents are highest in Southern Japan Trench (Fig. 5C)” but figure 5C represents Total carbon. Please rephrase. Fig. 5D *Artichnus* should be in italics.

Corrected that by replacing “C and N” by “total C” (line 219).

Thank you!

Response to Reviewers

Reviewer #2 (Remarks to the Author):

I am pleased to read this latest version of the manuscript. I believe that the discussion is clearly supported by the data now. Thus, I think is suitable for publication. Well done.

Minor.

Be careful in Fig. 5.

Ichnogenera should be in italics in some of the legend plots. For example, Fig. 5D and F *Artichnus* should be in italics. Same for Fig. 5F ichnogenera.

Fig. 5 F should be on the left bottom corner considering alphabetical order. Or changing the order of the plots; now it seems a bit confusing.

Trace fossil names are in italics now and the order of the plots is changed as suggested.
Thank you for the comments.